# Optimization of the System of Allocation of Overdue Loans in a Sub-Saharan Africa Microfinance Institution †

**Andreia Araújo** [1,2,*] **, Filipe Portela** [1,*] **, Filipe Alvelos** [1] **and Saulo Ruiz** [2]

1   Algoritmi Centre, University of Minho, 4800-058 Guimaraes, Portugal; falvelos@dps.uminho.pt
2   Pelican Rhythms, 4100-456 Porto, Portugal; saulo@pelican-technology.com
*   Correspondence: a75270@alunos.uminho.pt (A.A.); cfp@dsi.uminho.pt (F.P.)
†   This paper is an extended version of our paper published in The 8th International Conference on Mining Intelligence & Knowledge Exploration, Hammamet, Tunisia, 1–3 November 2021.

**Abstract:** In microfinance, with more loans, there is a high risk of increasing overdue loans by overloading the resources available to take actions on the repayment. So, three experiments were conducted to search for a distribution of the loans through the officers available to maximize the probability of recovery. Firstly, the relation between the loan and some characteristics of the officers was analyzed. The results were not that strong with F1 scores between 0 and 0.74, with a lot of variation in the scores of the good predictions. Secondly, the loan is classified as paid/unpaid based on what prediction could result of the analysis of the characteristics of the loan. The Support Vector Machine had potential to be a solution with a F1 score average of 0.625; however, when predicting the unpaid loans, it showed to be random with a score of 0.55. Finally, the experiment focused on segmentation of the overdue loans in different groups, from where it would be possible to know their prioritization. The visualization of three clusters in the data was clear through Principal Component Analysis. To reinforce this good visualization, the final silhouette score was 0.194, which reflects that is a model that can be trusted. This way, an implementation of clustering loans into three groups, and a respective prioritization scale would be the best strategy to organize and assign the loans to maximize recovery.

**Keywords:** assignment problem; data mining; microfinance

## 1. Introduction

This work is based on an after-stage from the usual loan process. The usual loan process should finish on the thirtieth day after the application was approved. Sometimes, this process can finish before that, and the loan is repaid earlier than the due date (which does not bring any question or problem to the company), but other times the process can take more than usual when the user is not capable of paying on time, and they enter into the overdue phase. This represents loss and a problem to the microfinance institution (MFI).

That overdue phase (before defaulting) can be divided into different stages and each one necessitates different approaches to the user. The problem approached in this study is the large amount of calls that officers need to make to the users that have an overdue loan.

Since there is not an allocation rule of the loans/lists to the officers and the rules that were presented previously are based on a business perspective, the company decided to search its data deeply to try to find a more technological approach to the question in order to optimize the lists.

So, the objective of this project started to be the development of a model of allocation between overdue loans and officers from the MFI Credit Collection teams (CC teams); the main research question is the optimization of the allocation of overdue loans to the CC team. With that in mind, three experiments were conducted to seek a distribution of the loans through the officers available to maximize the probability of recovery. The first two were based on the classification technique of data mining, and the last uses the clustering technique, which achieved the following:

- Definition of a prioritization scale to call the users with overdue loans;
- Definition of the list of contacts of the officers based on the prioritization scale.

This article extends a previous work that only presented one approach to this problem, the clustering one, by adding the two approaches related to the classification techniques [1]. Consequently, it presents more details about the processes and their results.

In addition, this work represents not only a scientific challenge but also keeps socially impacting the development of emerging markets. This is important to refer to as it follows the MFI vision of "Impacting people's lives in emergent markets through technology and entrepreneurship".

Lastly, to structure the communication of this project into this paper, it was decided to divide it into seven sections: Introduction, Background (where the context of the project is presented), Materials and Methods (where some tools are described), Case Study (where it is approached a data characterization and how it was modeled), Results and Discussion, and Conclusion.

## 2. Background

In this section, a context of the problem is presented and it is described why microfinance is important and how a loan application proceeds.

### 2.1. Importance of Microfinance

During the last global recession faced in 2008, one of the systems that struggled was the banking system. In a country, this has a negative impact on its economy and on its development. In Sub-Saharan Africa, the story was not different, and some countries of the region are still facing economic recovery from their latest recessions, reflecting a lot of challenges to the stability of the banking system. Adeyemi [2] addresses three variables as a big part of the failure in reaching that stability:

- Nonperforming loans
- Capital inadequacy
- Non-transparency

The focus of this work was only on the first point, which Reinhart et al. [3] refers to as a major cause of the economic failure or even the start of a banking crisis. Nonperforming loans are overdue loans by at least 90 days from the considered due date ([4]).

To fight poverty and benefit the social balance between economic classes, online microfinance was taken as a solution ([5]). Peer-to-peer (P2P) is one of the modalities created and it is responsible for making the connection between private lenders and borrowers in developing countries, through the internet ([6,7]), offering to entrepreneurs the opportunity to have access to a quicker option of credit.

### 2.2. Process of an Application to Loan

For the Sub-Saharan MFI under study, the investors can invest in the MFI aiming to grow a loan portfolio, but they will not decide where loan their money goes. This is made automatically when the application is processed and accepted, so the borrowers will receive the amount the moment there is a final decision. The lending process can be defined in five steps:

1. The user must download the app and log in;
2. The user fills in the profile and loan details;
3. The loans are evaluated based on the probability of default by processing all the details gathered from the device;
4. If the loan is accepted, the money is paid out to the bank account from the user directly;
5. The user repays via bank transfer or online transaction using a debit card, within an interval of 30 days after the loans get accepted.

This last step can be extended if the user does not pay on time. The user is granted a grace period of 3 days after the due date and if afterward it still has not been paid, the loan is redirected to overdue where the credit collection team will contact the user via SMS, app notifications, email, or calls to make them aware of the payment and the risks associated.

### 2.3. Related Works

Related to the microfinance area, the works found are more about credit scoring. This type of work, even if they are not the same topic as this present work, is helpful in the sense of what features are being selected to evaluate a loan, since in a way the output expected from the present work is a revaluation of the loan and if it will be overdue or not. For that reason, the features selected for each data set of the present work were based on the ones referred to in credit scoring models, with the time range of their collection being the only difference for some attributes during the loan payment period (from approval to due date). Some of the features usually used in credit scoring studies are personal information, housing, job title, or loan reason, as can be seen in Huang et al. [8] or West [9].

This article is an extension of previous work that only focused on the clustering technique to solve this problem. This technique was not the only one tested since it followed, as mentioned before, the credit scoring approach as a basis, so the first step would go to build and test classification techniques. Therefore, the work presented in Araújo et al. [1] is used as a support for the present article.

## 3. Materials and Methods

In this section, the auxiliary techniques used will be described that do not contribute directly to the final result but that impact the way the results could be achieved.

### 3.1. Pratical Method

When starting the research project, it is important to define the most suitable approach and philosophy in order to better formulate the research plan for the research question of how to optimize the allocation of overdue loans to CC team officers.

In order to define the artifact, a data mining process will be done. So, for this reason, a cross-industry standard process for data mining (CRISP-DM) methodology can be followed, which describes an approach to data mining [10].

Since the results demanded to take steps back on the process in order to make it possible to achieve a plausible solution, a CRISP-DM process adaptation to this project was done to complete three iterations. These iterations can be found in Figure 1.

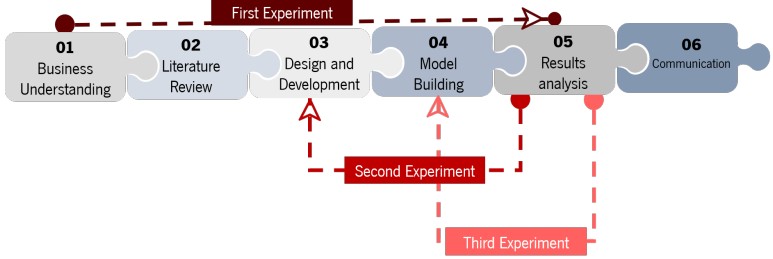

**Figure 1.** CRISP-DM process behaviour in practise of this work.

In the first experiment, the path designed was to fulfill all the steps from the first to the sixth. However, since the results were not applicable, there was a need to go back two steps and redesign a new approach—the second experiment.

In the second experiment, the first two steps were saved due to the problem characterization and company background being the same. The difference laid in the structure and collection of new features, and consequently a new dataset was extracted. Unluckily, the results were similar to the first experiment, which would mean a need for a new iteration—the third experiment.

Finally, in the third experiment, since it used the same dataset as in the second experiment and that allowed to already have enough knowledge about the features, it was only needed to go back to the model building phase.

*3.2. Tools*

SQL programming language was used to access the databases and to build the tables with the features for each experiment to be extracted. After extraction, the data were prepared by to Python programming language on Jupyter Notebooks and, consequently, analyzed too. In order to be able to proceed with the analysis, visualize the patterns, and predict the data in this language, a couple of libraries needed to be enabled:

- Pandas: it provides a structure for the data allowing to manipulate it by columns ([11]);
- Matplotlib: it is the most used library to plot graphs and other 2D data visualization ([11]);
- Seaborn: it is a library that provides data visualization, but with more plot styles and color options ([12]);
- Scikit-learn: it has the majority of the machine learning algorithms from techniques such as classification, regression, or clustering; but it also has some important tools such as the principal component analysis or the metrics used to evaluate the models ([13]);
- Kneed: to make it possible to identify easily the number of clusters in a dataset, this library has the KneeLocator, making it automatic ([14]).

*3.3. Algorithm Selection*

Firstly, in Huang et al. [8], support vector machine (SVM) presents some competitive results, compared to other algorithms, and it shows the same accuracy as it is described academically for credit scoring models. For the different experiments this study will report, SVM will be used without the mentioned combination, and decided after if the results could be improved recurring to the improvement of this technique.

Second, in Byanjankar et al. [15] neural networks showed successful results when classifying paid or defaulted loans and predicting the risk of the credit investment. Another positive point of this paper is that the features used in modeling are quite similar to the ones used in the present study.

Third, Paleologo et al. [16] exemplifies the importance of robust models when there is the case of unbalanced data, but it also shows in the results that decision trees can have good achievements when applied to credit scoring, so this algorithm will be present as well to compare outputs.

Fourth, classification models are also used in customer relationship management in order to understand and to induce the customer behavior as can be conferred in Ngai et al. [17]. This is important to refer to since it can support the credit scoring models, where the objective is to find the payment behavior of the user applied. In Ngai et al. [17], it is possible to see some of the algorithms already mentioned, among others. It was decided to use Naive Bayes to compare the outputs of this study based not only on what Ngai et al. [17] reviewed but also in the results concluded by Baesens et al. [18], where Naive Bayes is stated as an interesting option to estimate the customer lifecycle. Even if the customer lifecycle is treated differently to the credit scoring problem, it can be interesting to see the results when applied to the concrete challenge this described project is facing.

Cheng et al. [19] show good results when K-means is applied in a situation of customer segmentation. This is an approach that can be adapted to the context of the problem that is being solved, so K-means will be the other algorithm selected to discover if a clustering approach to prioritize the loans called and then allocate them to the officers would be the most beneficial.

### 3.4. Feature Encoding

The feature encoding can be explained simply as the transformations needed for the data to be translated in a numerical way that a machine learning algorithm can use as an input.

In this subsection, only the set of methods important for this study will be presented:

- Standard Scaler: Divides the value by a *normalization constant*. This does not change the shape of each feature distribution but normalizes the values to be understood by the model ([20]);
- Dummy Codding: This is one of the methods to encode categorical variables (qualitative variables). It transforms labels into a new feature with 0 or 1 and it guarantees an advantage when compared to the one-hot encoder (another method to encode categorical variables)—it removes one degree of freedom and it uses one less feature in the representation ([20]);
- Label Encoder: It is also used to encode categorical variables, with multiple labels. It associates with these labels one different and progressive integer number ([21]).

### 3.5. Confusion Matrix

In the case of classification algorithms, there is a Confusion Matrix that is created for each model. The Confusion Matrix shows four different results:

- True Positive (TP): when the model predicts correctly the positive value, in this case the variable *paid* is 1;
- True Negative (TN): when the model predicts correctly the negative value, in this case the variable *paid* is 0;
- False Positive (FP): when the model predicts wrongly the positive value, in this case the model returns variable *paid* as 0 but in reality it is 1;
- False Negative (FN): when the model predicts wrongly the negative value, in this case the model returns variable *paid* as 1 but in reality it is 0.

### 3.6. Classification Report

Based on the confusion matrix, three metrics evaluate the quality of a model to predict the reality, which is part of the classification report:

- Precision: is 1 when the model does not label a positive value as negative. So, this represents the percentage of positive values that were positive (positive prediction). It is calculated as:

$$Precision = \frac{TP}{TP + FP} \tag{1}$$

- Recall: is 1 when the model recognizes all the positive instances. This represents a fraction of positive values that were well identified. It is calculated as:

$$Recall = \frac{TP}{TP + FN} \tag{2}$$

- F1 Score: is the balance between Precision and Recall. It is calculated as:

$$F1Score = 2 \times \frac{(Recall \times Precision)}{(Recall + Precision)} \tag{3}$$

These measures will be used further to evaluate each algorithm from the first two experiments.

### 3.7. Elbow Method

The elbow method is a heuristic used in K-means to discover the best possible number of clusters a particular dataset can be divided into ([13]). It simply iterates a range of values $K$ (number of clusters), and for each iteration $K$ it calculates the sum of squared errors (SSE).

The purpose is to minimize this SSE, but since it decreases to 0 as $K$ is increased, the best solution is when the SSE is minimum for a low $K$. Otherwise, it is possible to have an SSE of 0 but that would mean all the data points would have their cluster and that is not what is meant to be achieved. So, what is expected to be found is a balance between the SSE and $K$, visually represented as the "elbow" of the graph, where the SSE starts decreasing at a constant pace.

### 3.8. Principal Component Analysis

Since large datasets are quite difficult to explore or visualize, the process of analyzing them and modeling becomes more difficult and less flowed. For this reason, sometimes, reducing the accuracy in the data slightly brings more benefits to be possible to achieve some meaningful outputs.

To show the potential of this method, as an example, Caruso et al. [22] used it to complement the analysis of regional economic activity in central Italy and to make it possible to achieve different conclusions, through visualization.

The aim when applying the principal component analysis (PCA) method is to reduce the dimensions of a large dataset without disturbing the majority of the information contained. It is possible to do this by transforming the big set of variables into a smaller one, always optimizing the preservation of its variability.

For this reason, during the process of finding new variables, it is settled that the variability needs to be maximized to make sure that it does not decrease from the initial value. To respect this goal, the new variables created are linear functions from the ones before, without correlating them.

### 4. Case Study

In this section, we describe the datasets used in the approach of the analysis presented in the paper and the modeling part.

### 4.1. Data Structure

Since the first approach was the prediction of what could be the best characteristics of the officers to deal with a specific loan and then improve the chance of recovering it, the variables needed to be more oriented to what could have a greater impact on this target and create the best allocation. The dataset used to model a solution for this experimentation, called dataset A from now on, considered the loans that went overdue and it was filtered not based on a period but on the sample of officers, selected for this study, that was the most recent contact of the specific loan, in order to increase the sample size.

Regarding the second and third approaches, they are more focused on the loan itself and the behavior of their users, independently of the characteristics of officers. On the second approach, the target was predicting the probability of payment and for the third, the target was clustering the loans. For these reasons, the dataset used to model both situations, which will be from here on referred to as dataset B, is similar to dataset A in terms of the loan-related variables that compose it, although the only difference is an increase in the number of SMS features extracted. Due to the large amount of time a query could take to run results of extraction, a need of restricting the period of the application of the loans arose. So, dataset B includes overdue loans that were granted in a period between 1 January 2019 and 1 November 2019.

Table 1 shows the principal differences between the two datasets, manifesting what needed to be improved from one to another in order to test the last two experiments.

**Table 1.** Description of an overview of dataset A and B, and its main differences.

|  | Dataset A | Dataset B |
|---|---|---|
| **Number of rows** | 46,132 | 26,825 |
| **Period** | Not defined | 1 January 2019 to 1 November 2019 |
| **Percentage Paid** | 54.46% | 55.50% |
| **Percentage Defaulting** | 45.54% | 44.50% |
| **Total number of variables** | 43 | 56 |
| **Features related to officers** | 9 | 0 |
| **Features related to SMS** | 10 | 32 |

Since both datasets were quite balanced, there was no need to apply any resampling strategy. The next subsections can be consulted for more details regarding the variables.

*4.2. Data Description*

As it was mentioned before, the variables collected follow the same basis as in credit scoring models, being related to personal information, loan characteristics ([8,9]) and mobile phone usage features ([23]).

Firstly, it is possible to access the personal information of a user from the moment a loan application is settled for the first time. There, the user needs to fill their profile to be able to interact with the service provided by this MFI. This profile includes personal information, such as date of birth and professional/educational background; it includes information related to their family (marital status and number of dependents) and property such as owning a car. Some of these variables can be updated later without creating a new profile to continue to apply for loans in this institution.

Secondly, the loan characteristics are added based on the purpose of each application a user makes. In this case, the amount and the reason for the loan are selected as well when filling different applications.

Finally, regarding the mobile phone usage features, two different approaches were taken for dataset A and dataset B. For dataset A, we used some features related to the installation of applications (app) and SMS received during the loan period (which is one of the differences for a credit scoring model), while on dataset B the features related to SMS were extended, not only with new variables but also collecting information from before the approval of the loan (adding to the loan period features previously on dataset A).

In addition, in dataset A, a sample of officers was set and some personal information was collected using an online form. The variables collected were related to demographics, educational background, and experience in the company measured by the number of working days, plus two other features that will be kept unknown to protect the personal profile of each officer. This sample worked as a restriction for what loans were considered in dataset A—the loans in which the last call to the user was made by one of these officers.

In Tables A1 and A2, in Appendixes A and B, it is possible to see the variables definition and their types in more detail for dataset A and B with some variables renamed to keep the confidentiality of this MFI.

In both Tables 2 and 3, there is a first description of some particular features to describe each datasets. This is a representation before data cleaning and it is already possible to identify where there is a lack of quality in the data, such as in the range of the variable age.

Note that for the protection of the information integrity of this company, the values presented are not absolute but they can show how the data behaved on this study and how the output can impact this MFI.

**Table 2.** Table of descriptive statistics for dataset A.

| | user_ age | marital_ status | nr_ children | employ_ status | educ_ status | officer_ age |
|---|---|---|---|---|---|---|
| | | | | **Dataset A** | | |
| Mean | 34.53 | 0.61 | 1.52 | 0.54 | 1.80 | 26.19 |
| Std dev | 14.03 | 0.56 | 1.87 | 0.64 | 0.41 | 3.28 |
| Min | 18.00 | 0.00 | 0.00 | 0.00 | 0.00 | 21.00 |
| 1st Q | 29.00 | 0.00 | 0.00 | 0.00 | 2.00 | 25.00 |
| 2nd Q | 34.00 | 1.00 | 1.00 | 0.00 | 2.00 | 26.00 |
| 3rd Q | 39.00 | 1.00 | 2.00 | 1.00 | 2.00 | 27.00 |
| Max | 1822.00 | 3.00 | 100.00 | 4.00 | 2.00 | 35.00 |

**Table 3.** Table of descriptive statistics for dataset B.

| | user_ age | marital_ status | nr_ children | employ_ status | education_ status |
|---|---|---|---|---|---|
| | | | **Dataset B** | | |
| Mean | 35.01 | 0.62 | 1.55 | 0.79 | 1.81 |
| Std dev | 11.81 | 0.56 | 1.84 | 1.16 | 0.40 |
| Min | 18.00 | 0.00 | 0.00 | 0.00 | 0.00 |
| 1st Q | 30.00 | 0.00 | 0.00 | 0.00 | 2.00 |
| 2nd Q | 34.00 | 1.00 | 1.00 | 1.00 | 2.00 |
| 3rd Q | 40.00 | 1.00 | 2.00 | 1.00 | 2.00 |
| Max | 1024.00 | 3.00 | 100.00 | 7.00 | 2.00 |

*4.3. Modeling*

In this modeling phase, it is possible to describe two different steps—model development and model validation. In the model development, it is expected to build the scenarios for each experiment and to explore the data to identify patterns and relationships to make the best prediction possible. Furthermore, the model validation will be when the prediction is tested and it is understood if it behaves correctly or not.

4.3.1. First Experiment

From the eight variables of dataset A (the features are described in Table A1 in Appendix A) collected to characterize an officer, only four of them will be considered to build a group of different models based on three of the main characteristics from the officers (two of the variables were merged).

For each of these three new variables responsible to characterize an officer, we decided to choose different dependent variables for modeling. In Table 4, the distribution of those variables can be found.

The logic behind this selection focused mostly on business reasons:

- **Demographic of the users:** would be allocated to the demographic variables of the officers;
- **Experience of the officers:** the *time_in_company* variable should be based on what stage the user is in if it is their first loan with this company, what type of loan it is, and the behavior of the users, here analyzed as SMS and app features;
- **Field of studies:** which should be related to with the academic background of the users and a mix of characteristics from the loan such as the amount, reason, or the bank entity where the loan was deposited.

**Table 4.** Distribution of the variables per different model.

| | Y | | |
| --- | --- | --- | --- |
| | **dem1Xdem2** | **field_of_studies** | **time_in_company** |
| **X** | user_age user_state user_gender marital_status nr_children owns_car | employment_status principal education_status reason entity_short_name | cardinal product_id sms_features (all) app_features (all) |

Since the desired output for the models is the probability of a certain characteristic of each of the three variables that build the profile of the officers to result in the best chance of payment from the user, it would be necessary to predict not only what characteristic would be optimal for each loan but also if it results in payment or not. For this, and already having dataset A divided into three subsets based on what was mentioned previously in Table 4, the subsets were filtered into new subsets where the three variables related to the officers would be constant. For example, the subset from the field of studies would be filtered by social sciences, natural/formal sciences, humanities, and applied sciences and professions, resulting in four new subsets. These subsets will be used in modeling to test and predict if the loan is paid or not. The characteristics that are being pursued are now input variables as well to predict the target variable *paid*, where a probability of payment will be associated with that particular characteristic. For example, the loan addressed to the field of study of social science would have a probability of 60% of payment.

All the subsets mentioned are trained and tested using SVM, neural networks, decision trees, and naive Bayes. The values from the metrics used to analyze the results should be between 0 and 1. The nearest to 1 the metrics are, the better a model can perform.

Note that dealing with personal characteristics opens the door of morals and ethics in the work and what is studied and evaluated during this project cannot call into question the discrimination of someone. So what was tested and analyzed during this project always took into consideration a barrier of not going too deep into the characteristics of the officers—if it did not work with the basics, the analysis related to the officers would not be carried out any further.

4.3.2. Second Experiment

After the conclusion of the first experiment, the best option would be to go back to the process and redesign a new approach to this problem. The new approach would focus, instead of on the best profile for an officer, on the probability of the loan being paid based on the loan characteristics only (considering just the behavior of the users in the respective loan, not including the company approaches to recover).

It was necessary to generate a new dataset, dataset B, with more features relating to the behavior of the users during the payment time for the loan—features relating to the SMSs and what could be inferred from their content. This dataset is described in Table A2 in Appendix B.

All the variables will be considered to build a unique model to predict the target variable–*paid*. This approach is the most similar to modeling credit scoring, which is the one used in the loan application. However, this time the trigger is not an application submission but a user not fulfilling the repayment on time, starting an overdue period with the risk of losing benefits in overall loan applications.

The target variable for a loan that came to be overdue is described as:

$$paid(x) = \begin{cases} 1 & \text{if the loan } x \text{ is recovered} \\ 0 & \text{if the loan } x \text{ is defaulted} \end{cases} \tag{4}$$

Going through with building a model that better fits the problem, the same models as in the first experiment are used for training and testing—SVM, neural networks, decision trees, and naive Bayes. The metrics that will serve as criteria to choose the best performing model are the ones used as well in the first experiment.

### 4.3.3. Third Experiment

This perspective followed was to try to find group patterns in the data already exported and prepared from dataset B (Table A2 in Appendix B) on the second experiment. This way, a clustering method (using K-means) was tested to discover if there are groups with similar behavior on the data and if it would be possible to prioritize the loans based on what group they would be part of.

The difference to the classification method is that in this case there is not an independent variable that is used to classify each loan. In this case, the target variable considered is a label that indicates which identified group the loan is part of, as mentioned in Araújo et al. [1]. The definition of these labels is based on the ideal number of clusters, identified here belonging to the elbow method ([13]). This method provides a $k$ number of clusters, which indicates the target variable varies between 0 and $k-1$, and it can be described mathematically as:

$$Cluster(x) = \begin{cases} 0 & \text{if the loan } x \text{ belongs to Cluster 0} \\ 1 & \text{if the loan } x \text{ belongs to Cluster 1} \\ \cdots \\ k-1 & \text{if the loan } x \text{ belongs to Cluster } k-1 \end{cases} \tag{5}$$

This value $k$ was calculated through the elbow method. In Figure 2, how the sum of squared errors behaves when the number of clusters increases is displayed.

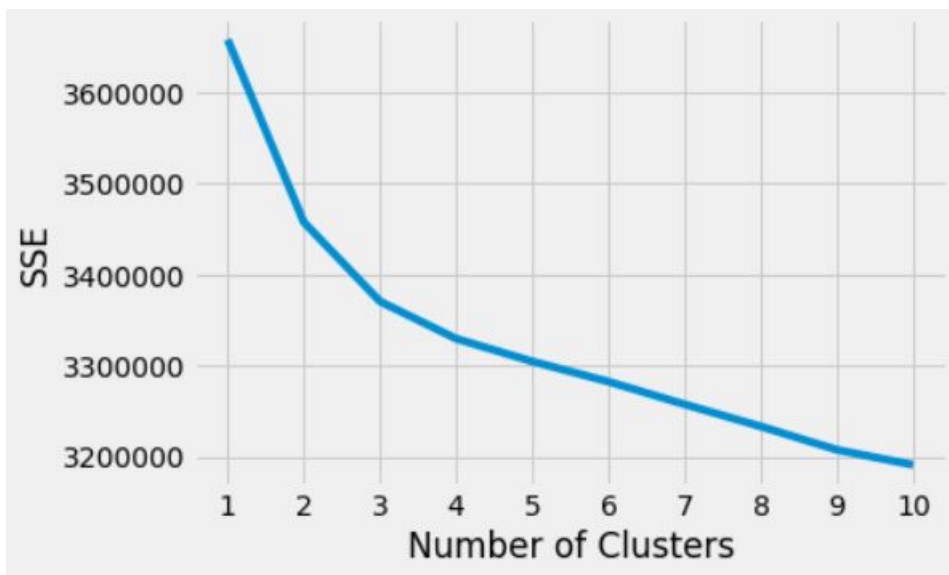

**Figure 2.** Optimal number of clusters displayed by the elbow method [1].

From the graphic in this Figure 2, the point is expected to be found where the balance between the sum squared errors and the number of clusters is optimal—the "elbow" of the curve. This point is when the number of clusters is 3, so from now on the aim is to visualize them on the data and characterize them.

## 5. Results and Discussion

In this section, the results are presented, explained, and analyzed based on the visualization of the clusters using PCA.

### 5.1. First Experiment

In Figures 3–5, respectively, representing the three different characteristics that build an officer profile—demographics merge, academic background, and experience in the company, how the average of the metric F1 score behaves is displayed along with the different subsets for each algorithm. The average comes from the F1 score results of predicting paid or not paid (1 or 0 from the target), respectively. Moreover, the decision of displaying the F1 score to summarize the results is related to the fact that it represents the balance between the other two metrics.

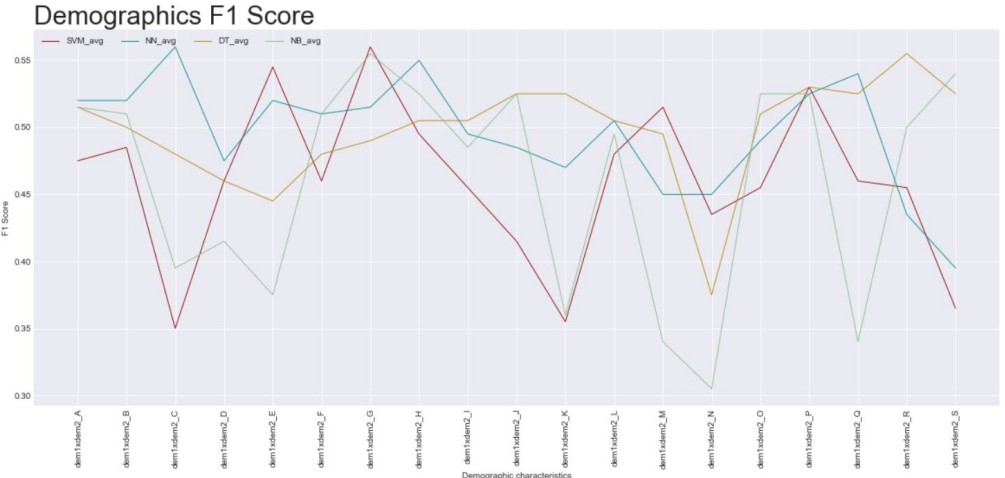

**Figure 3.** Variation of the F1 Score along the different tables related to variable *dem1Xdem2* modelled and tested.

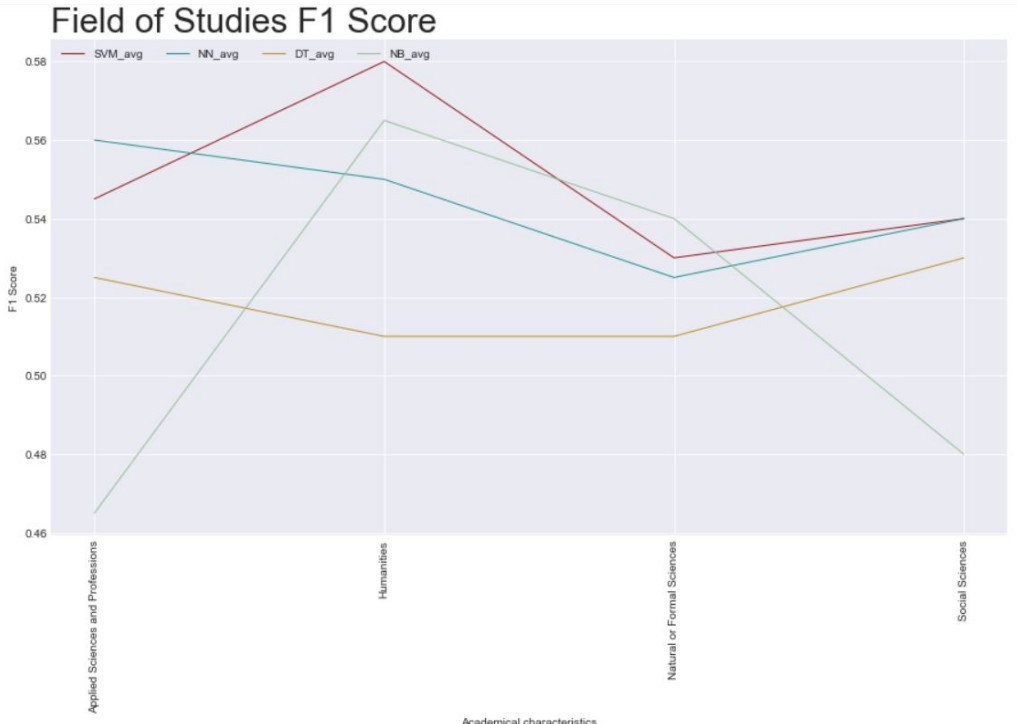

**Figure 4.** Variation of the F1 Score along the different tables related to variable *field_of_studies* modelled and tested.

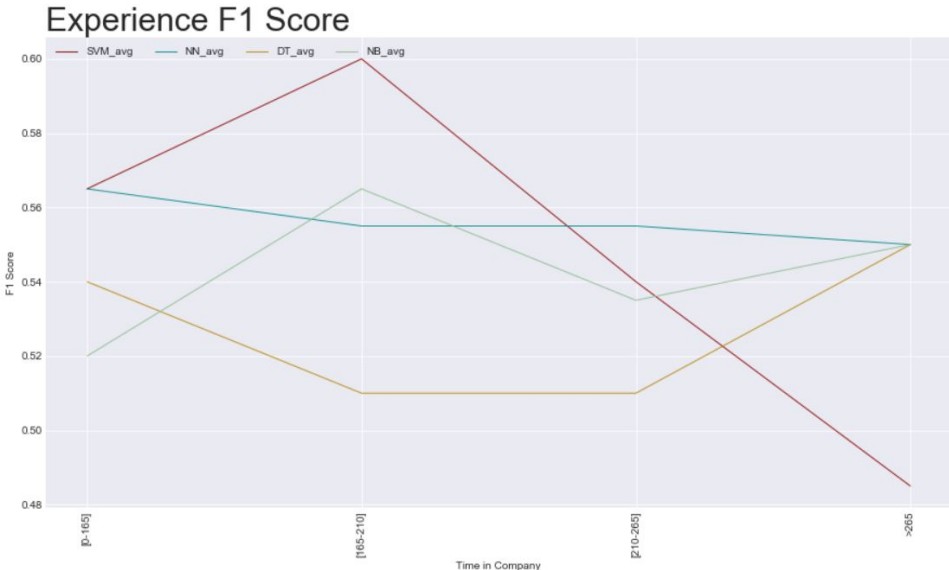

**Figure 5.** Variation of the F1 Score along the different tables related to variable *time_in_company* modelled and tested.

Regarding the demographics (Figure 3), only three subsets could achieve an average of over 0.55, which is not a great value by itself. In the academic background (Figure 4), the value is relatively better, as it reaches 0.58, and the values from each subset have less difference between them; however, it is not enough to prove to have a valuable model. Lastly, regarding the experience of the officers in the company (Figure 5), the average value is also slightly better when compared to the other characteristics—0.60—but the algorithm that represents the highest average is also the one that represents the lowest one, so it shows a lot of variation and a lack of confidence in this output as well.

To sum up, overall, the results were weak, which makes it possible to conclude that any of the algorithms tested in this experiment would behave quite randomly if applied.

### 5.2. Second Experiment

In Figure 6, it is possible to see the F1 score results from each algorithm tested from the result of each target variable to their mean.

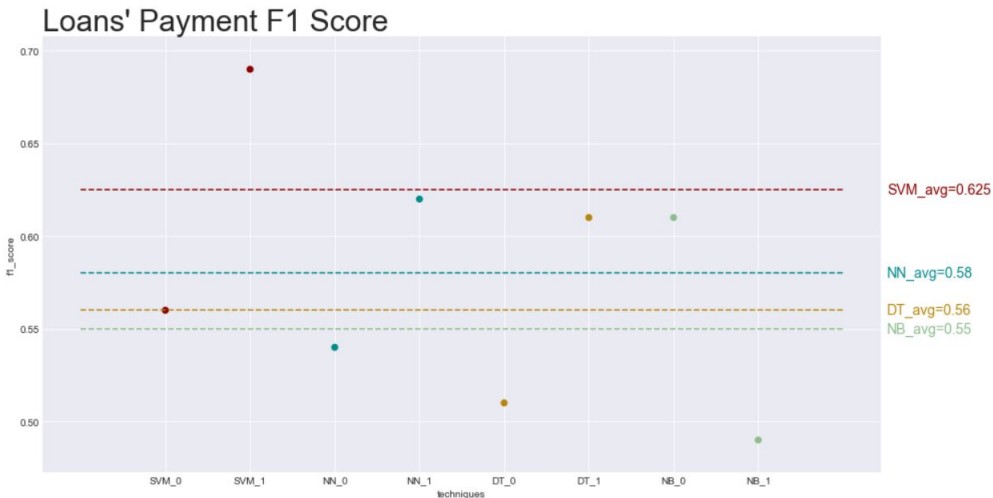

**Figure 6.** Variation of the F1 Score in the different algorithms tested.

The SVM algorithm presents a potentially good result, with an average of 0.625, but there is something that must be taken into consideration—the good prediction is made for

the paid loans, represented by the point *SVM_1* on the graph with 0.70. When looking at the prediction of unpaid loans (*SVM_0*), the F1 score falls to 0.55, which will behave as a random attribution of unpaid (just as flipping a coin, the loan can be labeled correctly as potentially unpaid the same number of times as wrongly). This will bring uncertainty to the way the officers would be organized, so it was decided not to implement it.

Furthermore, the rest of the results found are weak, very similar to the first experiment, which is not enough to implement a model—it would end up resulting in a random allocation. So, for what was tested in the first and second experiments, a classification approach for a solution does not have satisfactory results. That is why a different approach was also considered as a third experiment.

### 5.3. Third Experiment

Succeeding the two new equivalent variables to deploy from the application of the PCA, the graph in Figure 7 was plotted, as mentioned in [1].

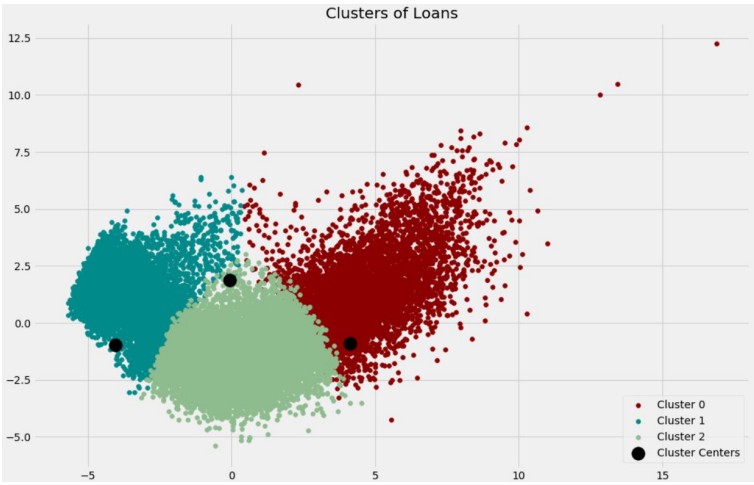

**Figure 7.** Visual representation by the PCA method of the three clusters present in this data [1].

From the analysis of the centers of each cluster, it is possible to understand the features that separate and organize each cluster from the others. The majority of the features are common so it was decided to resample the initial data containing just the features where it is possible to see big oscillations in the coordinates of the centers of the cluster. Furthermore, the silhouette score was only 0.043, which even if it is higher than 0, is still poor. The reason it is that low could be related to the number of features in common in the analysis of the centers.

The new subset contains all the variables related to the SMSs and the variable *principal*, which is the amount of money requested by the user.

Therefore, the modeling process was repeated and a new graph of the clusters was plotted. The graph is shown in Figure 8 [1].

Note that since the K-means model was redone, the legend from Figure 8 is different as the new number labeled is settled randomly, so Cluster 1 from Figure 7 is not the same as Cluster 1 from Figure 8. The comparison between both representations will be based on the size and the position of the points in each cluster.

Consequently, there are minimal differences between the previous graphic and this new one, which can verify that these features are the ones that impact the cluster formation. Furthermore, the silhouette score also reflected it, with an improvement to 0.194. Since this score varies between −1 and 1, in practice 0.194 would mean trust in the model.

From a business perspective, for the company case, created three criteria were created to define which clusters would have the loans that require higher prioritization. These criteria can be described, with the order mattering, as:

1. **Percentage of loans recovered per cluster:** To check where the amount of paid loans is the highest, and consequently with more probability of a return;
2. **Amount of defaulting money per cluster:** To show where the biggest amount of money can be lost is. This results from the sum of the variable *principal*;
3. **Average from a particular SMS feature:** This criterion will not be described in detail to protect the business side of this MFI.

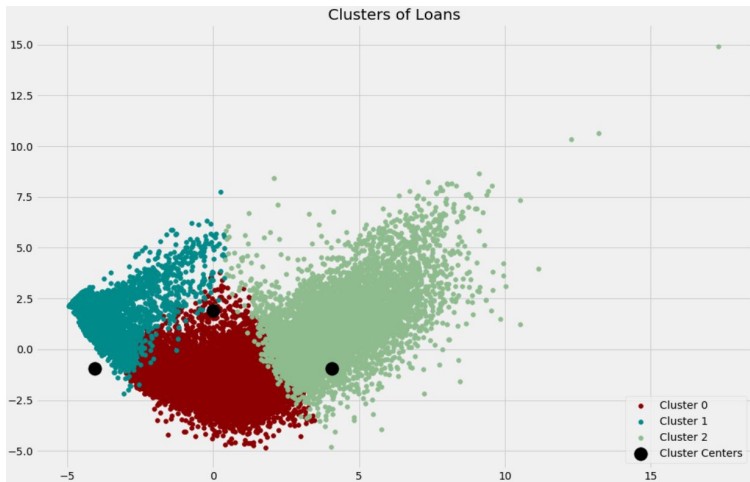

**Figure 8.** Visual representation by the PCA method of the three clusters present in the resample after analyse the centers [1].

After setting these criteria, a table was built, Table 5, where the results for each cluster can be consulted. Furthermore, these were analyzed following the importance of each criterion (shown by the order enumerated).

**Table 5.** Criteria results per each cluster [1].

| Cluster | Percentage of Loans Recovered | Amount of Defaulting | Average of an SMS Feature |
|---|---|---|---|
| **0** | 51.77% | 124,178,220.00 | 59,024,711.00 |
| **1** | 50.50% | 66,444,430.00 | 70,482.00 |
| **2** | 67.77% | 89,303,870.00 | 94,007,441.00 |

For the first criterion, Cluster 2 has the highest percentage whereas Cluster 0 and Cluster 1 are similar. So, there is now some advantage for Cluster 2 in the prioritization scale.

For the second criterion, Cluster 0 represents a higher amount of possible defaulting money, but Cluster 2 shows high values as well. For this reason, Cluster 2 keeps the advantage of being the most prioritized (since the first criterion has more weight on the decision), but Cluster 0 is now settled to be the second in the prioritization scale.

For the third and last criteria, the results came to confirm that Cluster 0 has more potential than Cluster 1.

To sum up, the prioritization for the loans should be in the following order:

1. Cluster 2
2. Cluster 0
3. Cluster 1

### 5.4. Allocation Model

After labeling the overdue loans, what is still missing is to allocate them to the officers available, following the prioritization rules. In order for that to be possible, an integer programming model was created to optimize the number of loans per officer. This was also presented in [1].

The first action is transforming the output from the clustering model into an input for this model. Thereunto, three variables were created to provide such information:

$$W_i = \begin{cases} 1 & \text{if the loan } i \text{ is part of Cluster 0} \\ 0 & \text{otherwise} \end{cases}$$

$$Y_i = \begin{cases} 1 & \text{if the loan } i \text{ is part of Cluster 1} \\ 0 & \text{otherwise} \end{cases}$$

$$Z_i = \begin{cases} 1 & \text{if the loan } i \text{ is part of Cluster 2} \\ 0 & \text{otherwise} \end{cases}$$

In addition, there is one more piece of information that comes from the clustering model—the prioritization of the loans. So, three weights were created, which will be associated with each cluster to add the value matching this prioritization scale, where $\gamma$ (weight of the loans belonging to Cluster 2) $> \alpha$ (weight of the loans belonging to Cluster 0) $> \beta$ (weight of the loans belonging to Cluster 1).

The goal of this allocation model is to maximize the total weight of the loans assigned to an officer and the mathematical model is:

$$\text{Maximize} \quad \sum_{i=1}^{n} \sum_{j=1}^{m} (\alpha W_i + \beta Y_i + \gamma Z_i) x_{ij}$$

$$\text{Subject to} \quad \sum_{i=1}^{n} x_{ij} = 1, \qquad \text{for each loan } j$$

$$\sum_{j=1}^{m} x_{ij} \leq k, \qquad \text{for each officer } i$$

$$x_{ij} \in \{0, 1\}, \qquad \text{for } i = 1, \cdots, n \text{ and } j = 1, \cdots, m$$

where $x_{ij} = 1$ if loan $j$ is assigned to officer $i$, 0 if not. Firstly, the objective function indicates the maximization of the allocation of most priority loans to the officers. Then, the first set of constraints ensures that every loan is assigned to only one officer and the second set of constraints ensures that every officer does not have more assigned loans than their capacity.

## 6. Conclusions

In this section, the conclusions are described in final considerations, and in future work as the next steps that this study can follow.

### 6.1. Final Considerations

With the increase of the number of users requesting loans in this MFI, there is a higher risk associated with having loans going overdue, as it would require more human resources available to be capable of handling all the calls needed to influence people to repay.

This MFI decided to take action and revise strategies to analyze, reorganize and improve the performance of the credit collection team. The focus would fall on options to fight the lack of the organization when assigning a loan to officers since there were not many rules and a loan was handled by different officers, always through the different steps, without knowing if it would be profitable or not to invest time in that loan.

Firstly, the idea was to study relationships between the characteristics of the loan and the officers, in order to build an officer profile capable of optimizing the connection with the user, to improve their management and have greater influence over them. Some data were collected and analyzed, and three different models (demographics, academic background, and experience) were built based on that, applying the classification methods of data mining. The results were not meaningful, since the F1 scores were quite low:

- Demographics: only three subsets could achieve an average of 0.55 as the maximum;

- Academic background: the difference between subsets has not that much variance; however, the best value is an average of 0.58;
- Experience: this was the one having the highest value—0.60, the worst of it is that the model that represents this, is the same that has the low value of all in another subset, which deconstructs the credibility on it.

This shows that the models performed randomly without a trend that could be followed and implemented in the context of this business.

Secondly, after the not so positive results of the first approach, a new experiment was directed into the characteristics of the loans. This means that the previous characteristics of the officers were not considered afterward and a new dataset needed to be considered to look for different behaviors from the users that could indicate repayment or not during the overdue approach by the credit collection team. New features related to the SMS activity before applying to the loan and during the payment time were also extracted and included in this new dataset. Even if the results seemed to be promising with an average F1 score of 0.625 for the SVM algorithm, it was not that powerful and satisfactory—the SVM for paid predictions was good with a score of 0.70; however, the SVM for unpaid predictions was only 0.55, which shows a random behavior as in the outcomes of the first experiment. This shows that a classification approach could not be the solution expected.

Third and lastly, the decision was to experiment with applying a different Data Mining technique—clustering. This way, instead of predicting if the loan has chances of being or not being paid, the objective was to group the loans and to find a prioritization scale for these groups. In this case, the K-means algorithm was used, which had three clusters well defined by PCA and a final silhouette score of 0.194, which shows a good possibility to rely on it. This way, it was possible to build a prioritization scale: Group 2 > Group 0 > Group 1, based on the rearrangement of the overdue loans into three groups (Cluster 0, Cluster 1, and Cluster 2) from these results, that made this project achieve the purposed goals.

The achievement of a system capable of clustering automatically the loans in the exact time that they go overdue and then allocate them daily to a respective officer from the credit collection team based on the prioritization scale built allows this MFI to optimize the overdue process, fulfilling the objective that they were looking for with this project.

In addiction, this is one more enforcement of the use of data mining algorithms, specifically the clustering technique, in finance. However, it is also a step further where it includes a crossover between two areas—data mining and operational research, and it shows the good use of a variety of techniques to complement each other.

*6.2. Next Steps*

Despite the objectives of this work being achieved, there is always areas where things can be improved or retested.

One of the areas is to generate more data and to understand if the behavior of the classification techniques improves. However, this could be more difficult due to the pandemic. It would be hard to generate more data with a similar context to the one that was tested in this study since the way the world is working, its processes, and how the money is flowing is different and has adapted.

Another area that can be improved is the information transfer between officers that called the same loan. For this, it is necessary to ensure that the officers get the proper training so there is a standardized way of filling out the information in the information systems of this company. This would ensure that the information transmission occurs as smoothly as possible. Although this is a point that was not the focus of this study, it appeared as a problem during the generation and extraction of the data. So, a revision and analysis of the onboarding and training process of the officers are left as well as a suggestion for the future, considering that it can be an obstacle in future studies performed in the company.

**Author Contributions:** Investigation, A.A.; Resources, S.R.; Supervision, F.P. and F.A.; Writing—original draft, A.A.; Writing—review & editing, F.P. All authors have read and agreed to the published version of the manuscript.

**Funding:** This research received no external funding.

**Data Availability Statement:** Not applicable, the study does not report any data.

**Conflicts of Interest:** The authors declare no conflict of interest.

## Appendix A. Variables from Dataset A

**Table A1.** Variables from Dataset A.

| Variable | Type | Description |
|---|---|---|
| loan_id | Nominal | Loan ID |
| cardinal | Scale | Number of loans accepted in this MFI |
| product_id | Scale | Different modes to apply |
| principal | Scale | Total amount applied |
| reason | Nominal | Reason to ask for credit |
| user_age | Scale | Age of the user |
| user_state | Nominal | City where user lives |
| address_info_complete | Binary | All the address information filled? |
| user_gender | Binary | Gender of the user |
| user_employment_status | Scale | User is unemployed or employed or other |
| employment_info_complete | Binary | All the employment information filled? |
| user_marital_status | Scale | If the user is married, single, for example |
| user_number_children | Scale | How many children an user has |
| owns_car | Binary | If the user has or not a car |
| user_education_status | Scale | What education degree the user has |
| entity_short_name | Nominal | Bank entity of the user |
| sms_feature_01 | Scale | Feature related to SMS of the user |
| sms_feature_02 | Scale | Feature related to SMS of the user |
| sms_feature_03 | Scale | Feature related to SMS of the user |
| sms_feature_04 | Scale | Feature related to SMS of the user |
| sms_feature_05 | Scale | Feature related to SMS of the user |
| sms_feature_06 | Scale | Feature related to SMS of the user |
| sms_feature_07 | Scale | Feature related to SMS of the user |
| sms_feature_08 | Scale | Feature related to SMS of the user |
| sms_feature_09 | Scale | Feature related to SMS of the user |
| sms_feature_10 | Scale | Feature related to SMS of the user |
| loan_installed | Binary | Was APP related to Loan installed? |
| competitor_installed | Binary | Was APP of a Competitor installed? |
| bank_installed | Binary | Was APP of a registered Bank installed? |
| otherbank_installed | Binary | Was APP of other Banks installed? |
| cryptocurrency_installed | Binary | Was APP about Cryptocurrency installed? |
| invet_trade_installed | Binary | Was APP about Invest/Trading installed? |
| money_installed | Binary | Was APP about making money installed? |
| officer_id | Nominal | Officer ID |
| officer_age | Scale | Age of the officer (years) |
| officer_gender | Binary | Gender of the officer |
| officer_demographic_01 | Nominal | Variable related to demographics of the officer |
| officer_demographic_02 | Nominal | Variable related to demographics of the officer |
| officer_language | Nominal | Which languages they know |
| officer_education_status | Scale | Education degree of the officer |
| officer_studies | Nominal | Field of studies |
| time_company | Scale | Number of days working on the company |
| paid | Binary | 0 if not paid; 1 if paid |

## Appendix B. Variables from Dataset B

**Table A2.** Variables from Dataset B.

| Variable | Type | Description |
|---|---|---|
| loan_id | Nominal | Loan ID |
| cardinal | Scale | Number of loans accepted in this MFI |
| product_id | Scale | Different modes to apply |
| principal | Scale | Total amount applied |
| reason | Nominal | Reason to ask for credit |
| user_age | Scale | Age of the user |
| user_state | Nominal | City where user lives |
| address_info_complete | Binary | All the address information filled? |
| user_gender | Binary | Gender of the user |
| user_employment_status | Scale | User is unemployed or employed or other |
| employment_info_complete | Binary | All the employment information filled? |
| user_marital_status | Scale | If the user is married, single, for example |
| user_number_children | Scale | How many children an user has |
| owns_car | Binary | If the user has or not a car |
| user_education_status | Scale | What education degree the user has |
| entity_short_name | Nominal | Bank entity of the user |
| sms_feature_01 | Scale | Feature related to SMS of the user |
| sms_feature_02 | Scale | Feature related to SMS of the user |
| sms_feature_03 | Scale | Feature related to SMS of the user |
| sms_feature_04 | Scale | Feature related to SMS of the user |
| sms_feature_05 | Scale | Feature related to SMS of the user |
| sms_feature_06 | Scale | Feature related to SMS of the user |
| sms_feature_07 | Scale | Feature related to SMS of the user |
| sms_feature_08 | Scale | Feature related to SMS of the user |
| sms_feature_09 | Scale | Feature related to SMS of the user |
| sms_feature_10 | Scale | Feature related to SMS of the user |
| sms_feature_11 | Scale | Feature related to SMS of the user |
| sms_feature_12 | Scale | Feature related to SMS of the user |
| sms_feature_13 | Scale | Feature related to SMS of the user |
| sms_feature_14 | Scale | Feature related to SMS of the user |
| sms_feature_15 | Scale | Feature related to SMS of the user |
| sms_feature_16 | Scale | Feature related to SMS of the user |
| sms_feature_17 | Scale | Feature related to SMS of the user |
| sms_feature_18 | Scale | Feature related to SMS of the user |
| sms_feature_19 | Scale | Feature related to SMS of the user |
| sms_feature_20 | Scale | Feature related to SMS of the user |
| sms_feature_21 | Scale | Feature related to SMS of the user |
| sms_feature_22 | Scale | Feature related to SMS of the user |
| sms_feature_23 | Scale | Feature related to SMS of the user |
| sms_feature_24 | Scale | Feature related to SMS of the user |
| sms_feature_25 | Scale | Feature related to SMS of the user |
| sms_feature_26 | Scale | Feature related to SMS of the user |
| sms_feature_27 | Scale | Feature related to SMS of the user |
| sms_feature_28 | Scale | Feature related to SMS of the user |
| sms_feature_29 | Scale | Feature related to SMS of the user |
| sms_feature_30 | Scale | Feature related to SMS of the user |
| sms_feature_31 | Scale | Feature related to SMS of the user |
| sms_feature_32 | Scale | Feature related to SMS of the user |
| loan_installed | Binary | Was APP related to Loan installed? |
| competitor_installed | Binary | Was APP of a Competitor installed? |
| bank_installed | Binary | Was APP of a registered Bank installed? |
| otherbank_installed | Binary | Was APP of other Banks installed? |
| cryptocurrency_installed | Binary | Was APP about Crypto installed? |
| invet_trade_installed | Binary | Was APP about Invest/Trading installed? |
| money_installed | Binary | Was APP about making money installed? |
| paid | Binary | 0 if not paid; 1 if paid |

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
