# Peer review of "Optimization of the System of Allocation of Overdue Loans in a Sub-Saharan Africa Microfinance Institution"

_futureinternet, doi:10.3390/fi14060163_

Round 1

Reviewer 1 Report

The authors should provide a survey of similar work for predicting overdue loans.

The classification results are not very satisfied. F1 is about 0.6 in a binary problem. The authors should try to use ensembles such random forest and stacking.

Feature importance could be interesting for the most accurate learner.

I cannot understand the usefulness of the clustering in this case study. It would be interesting only in a cluster then label procedure.

Author Response

Dear reviewer, thank you for your review and help turn our article better.

Reviewer 1:

Regarding similar work about predicting loans, the title was revised. In the subsection Related Work, it was also made clear what was expected to know in the project as if the loan will be overdue or not and how we could reevaluate the loan. That’s why the similar works used are more related to credit scoring since they use the same basis of evaluation.

Regarding the points of not being clear that an F1=0.6 was not enough to be used, plus having one experiment related to clustering, We added a new paragraph on the discussion of the results of the second experiment: “Furthermore, the rest of the results found are weak, very similar to the first experiment, which is not enough to implement a model - it would end up resulting in a random allocation. So, for what was tested in the first and second experiments, a classification approach for a solution does not have satisfactory results. That is why a different approach was also considered a third experiment.” We rephrased some other parts of the discussion to have more focus on this.

Reviewer 2 Report

This paper focuses on the prediction of overdue allocation loans in Sub-Saharan Africa microfinance institutions.

The paper is interesting, and it is suitable for this journal. There are several strengths but also some issues to resolve.

  1. Please underline which are the main results.

  1. Literature review could be further improved. In particular, since the visualization of the clusters is obtained through Principal Component Analysis, I suggest considering the following work:

https://www.researchgate.net/publication/339108905_A_Micro-level_Analysis_of_Regional_Economic_Activity_Through_a_PCA_Approach

  1. Please highloght which is the novelty of your work.

I encourage the authors to refine their paper to make it available for publication in the journal.

Author Response

Dear reviewer, thank you for your review and help turn our article better.

About the first point to underline the main results, the conclusion section, more precisely in the Final Considerations, had some changes in order to highlight that the objective (better expressed with the new title) was fulfilled. Some parts about describing each experiment were cut to not confuse with the main results, and a paragraph to make them more objective was added: “The achievement of a system capable of clustering automatically the loans in the exact time that they go overdue and then allocate them daily to a respective officer from the Credit Collection team based on the prioritization scale built allows this MFI to optimize the overdue process, fulfilling the objective that they were looking for with this project.”

For the literature review part, there was a lack of mentioning a literature reference in the elbow method in the section Materials and Methods due to a distraction since it was appearing later in the article. The suggestion of adding that paper to the PCA review was well accepted (“To show the potential of this method, as an example, Caruso et al. used it to complement the analysis of the regional economic activity in central Italy and to make it possible to achieve different conclusions, through visualization.”).

Regarding the point of highlighting the novelty of the work, a new paragraph was added in the Introduction to make clear that doing this type of work in a real company of an emergent market has an impact on sharing knowledge between countries but also thrives for development (“In addition, this work represents not only a scientific challenge but also keeps impacting socially the development in emerging markets. This is important to refer to as it follows this MFI vision: “Impacting people’s lives in emergent markets through technology and entrepreneurship”.”).

Round 2

Reviewer 1 Report

Random Forest and Gradient Boosting should be also examined.

Author Response

"Random Forest and Gradient Boosting should be also examined" It will be included in future work, because it requires much time to execute it.